# Fully-automated and ultra-fast cell-type identification using specific marker combinations from single-cell transcriptomic data

Aleksandr Ianevski [1,2], Anil K. Giri [1✉] & Tero Aittokallio [1,2,3,4✉]

Identification of cell populations often relies on manual annotation of cell clusters using established marker genes. However, the selection of marker genes is a time-consuming process that may lead to sub-optimal annotations as the markers must be informative of both the individual cell clusters and various cell types present in the sample. Here, we developed a computational platform, ScType, which enables a fully-automated and ultra-fast cell-type identification based solely on a given scRNA-seq data, along with a comprehensive cell marker database as background information. Using six scRNA-seq datasets from various human and mouse tissues, we show how ScType provides unbiased and accurate cell type annotations by guaranteeing the specificity of positive and negative marker genes across cell clusters and cell types. We also demonstrate how ScType distinguishes between healthy and malignant cell populations, based on single-cell calling of single-nucleotide variants, making it a versatile tool for anticancer applications. The widely applicable method is deployed both as an interactive web-tool (https://sctype.app), and as an open-source R-package.

[1] Institute for Molecular Medicine Finland (FIMM), HiLIFE, University of Helsinki, Helsinki, Finland. [2] Helsinki Institute for Information Technology (HIIT), Aalto University, Helsinki, Finland. [3] Institute for Cancer Research, Department of Cancer Genetics, Oslo University Hospital, Oslo, Norway. [4] Centre for Biostatistics and Epidemiology (OCBE), Faculty of Medicine, University of Oslo, Oslo, Norway. ✉email: anil.kumar@helsinki.fi; tero.aittokallio@helsinki.fi

Accurate identification of distinct cell types in complex tissue samples is a critical prerequisite for elucidating the roles of cell populations in various biological processes including hematopoiesis, embryonic and intestinal development[1–4]. Traditionally, cell sorting and microscopic techniques have been extensively used to isolate cell types, followed by molecular profiling of the sorted cells using, for instance, mRNA or protein measurements[5–7]. Decades of research has led to several collections of cell-specific features, including expression of marker genes in various tissues that are being used to distinguish various cell types in new tissue samples[8,9]. However, the entire process is manually tedious and technically challenging. Recently, single-cell RNA sequencing (scRNA-seq) has been established as a high-throughput approach to routinely chart diverse cell populations in tissue samples and to study various biological processes in disease and development[2,10–12]. The scRNA-seq technology has provided an unprecedented view of various cell types and it has become the leading technology in large-scale cell mapping projects such as the Human Cell Atlas[13].

Identification of cell populations often relies on unsupervised clustering of cells based on their transcriptomic profiles, followed by cluster annotation using marker genes that are differentially-expressed between the clusters[14,15]. These marker genes are then manually inspected using available information in the literature or cell marker databases[8,9] to assign cell type labels to each detected cluster. However, the manual selection of cluster-specific marker genes is a time-consuming and error-prone task, since the marker genes are often (i) expressed in multiple cell clusters, and (ii) correspond to multiple cell types. In addition, the expression of negative marker genes, which provide evidence against a cell being of a particular type, should also be incorporated into the cell-type identification process. The cell annotation procedure is further complicated by the lack of curated cell marker databases that include both known and de novo positive and negative markers to annotate cell-types with confidence. For example, selection of *CD44* as marker gene may compromise the accuracy of cell annotation as *CD44* is expressed in various immune cell populations[8]. Another popular approach for cell-type assignment is to utilize a reference dataset, a collection of previously annotated cell types in single-cell data, to train a classification algorithm and to apply it to new single-cell datasets. However, such supervised approaches require that the reference and new datasets resemble each other, which often pose a problem in scRNA-seq studies[16].

One important application of single-cell characterization is to design personalized treatments that selectively target malignant cell types in a patient-derived sample, while avoiding severe inhibition and toxic effects on healthy cells[17]. In cancers and other complex diseases, monotherapy resistance often emerges and requires multi-drug co-inhibition of various disease- and resistance-driving cell populations. We recently demonstrated how our comprehensive ScType marker database helped an AI-guided identification of personalized drug combination therapies for patients with refractory acute myeloid leukemia (AML), which led to synergistic co-inhibition of leukemic cell subpopulations that emerged in various stages of the disease pathogenesis and treatment regimens[18]. These cancer-selective and patient-specific combinations were shown to be relatively less toxic to lymphoid cells (non-malignant cells in the AML case), thereby increasing their likelihood for clinical translation. However, how to accurately distinguish between multiple malignant and non-malignant cell populations for targeted treatments remains a translational challenge and requires both systematic and highly selective strategies that are applicable to various diseases and tissue types. In many biomedical applications, reference single-cell data and cell-type annotations are not available, rather the cell population identification needs to be done individually for each patient sample.

To solve these challenges, we developed a computational ScType platform (marker database and cell-type identification algorithm), which requires only a single scRNA-seq dataset for accurate and unsupervised cell-type annotation (Fig. 1a). The unbiased yet selective cell-type annotation is achieved by compiling the largest database of established cell-specific markers (ScType database), and by ensuring the specificity of marker genes across both the cell clusters and cell-types (ScType specificity score, see Fig. 1b, c). We carried out a systematic benchmarking of ScType and related methods across 6 scRNA-seq datasets from four human and two mouse tissues, and showed that ScType platform correctly annotated a total of 72 out of 73 cell-types (98.6% accuracy), including 8 newly-reannotated cell-types that were incorrectly or non-specifically annotated in the original studies. In addition, ScType implements a single-cell single-nucleotide variant (SNV) calling option to distinguish between malignant and non-malignant cells (Fig. 1a), exemplified here using scRNA-seq data from AML patient sample. This case study demonstrates how the ScType platform can be used for anticancer applications, such as data-driven identification of leukemic cell populations toward personalized and selective treatment selection. ScType platform is implemented as an open-source and interactive web-tool (https://sctype.app), connected to the ScType marker database, to enable ultra-fast and fully-automated cell-type annotation in a wide range of biomedical applications.

## Results

### ScType improves annotation of cell types based solely on a given scRNA-seq data.
We first investigated the performance of ScType by re-analyzing a published scRNA-seq study of human liver cells[10]. Using only the raw scRNA-seq data from the liver atlas dataset, ScType automatically identified 17 clusters and correctly assigned them to 11 identified liver-related cell types that were manually annotated in the original study (Fig. 1d). This demonstrates the benefits of the comprehensive marker databases and the accuracy of the fully-automated annotation process. Additionally, ScType was able to automatically distinguish between two closely-related cell populations of B-cells (immature and plasma B cells) that were not differentiated in the original manuscript[10]. This segregation between immature and plasma B cells was done based on the positive and negative information in the ScType database that plasma cells do not express common B-cell markers, such as *CD19* and *CD20*, but instead they express *CD138* (Fig. 1e)[19].

Next, we re-analyzed another published scRNA-seq data of mouse retinal cells (Supplementary Fig. 1a)[20]. ScType automatically identified three closely-related cell populations of amacrine cell types (GABAergic, glycinergic and startbust), which were originally-identified by an extensive and deep analysis of selectively-expressed markers[20]. Furthermore, ScType correctly distinguished between the two subtypes of bipolar cells–rod (expressing *PRKCA*[21] and *CAR8*[22],) and cone (expressing *SCGN*[23], Supplementary Fig. 1b) bipolar cells, which were manually assigned to a single group in the original study, therefore enhancing the resolution of the cell-type annotation. Taken together, these results indicate that ScType enables a fully-automated prioritization of highly-specific markers for accurate annotation of even rare cell-types with distinct and selective molecular features.

### Systematic benchmarking of ScType in human and mouse scRNA-seq data sets.
To investigate the wider applicability of the automated method, we next benchmarked the performance of ScType in terms of its ability to automatically assign cell-types in

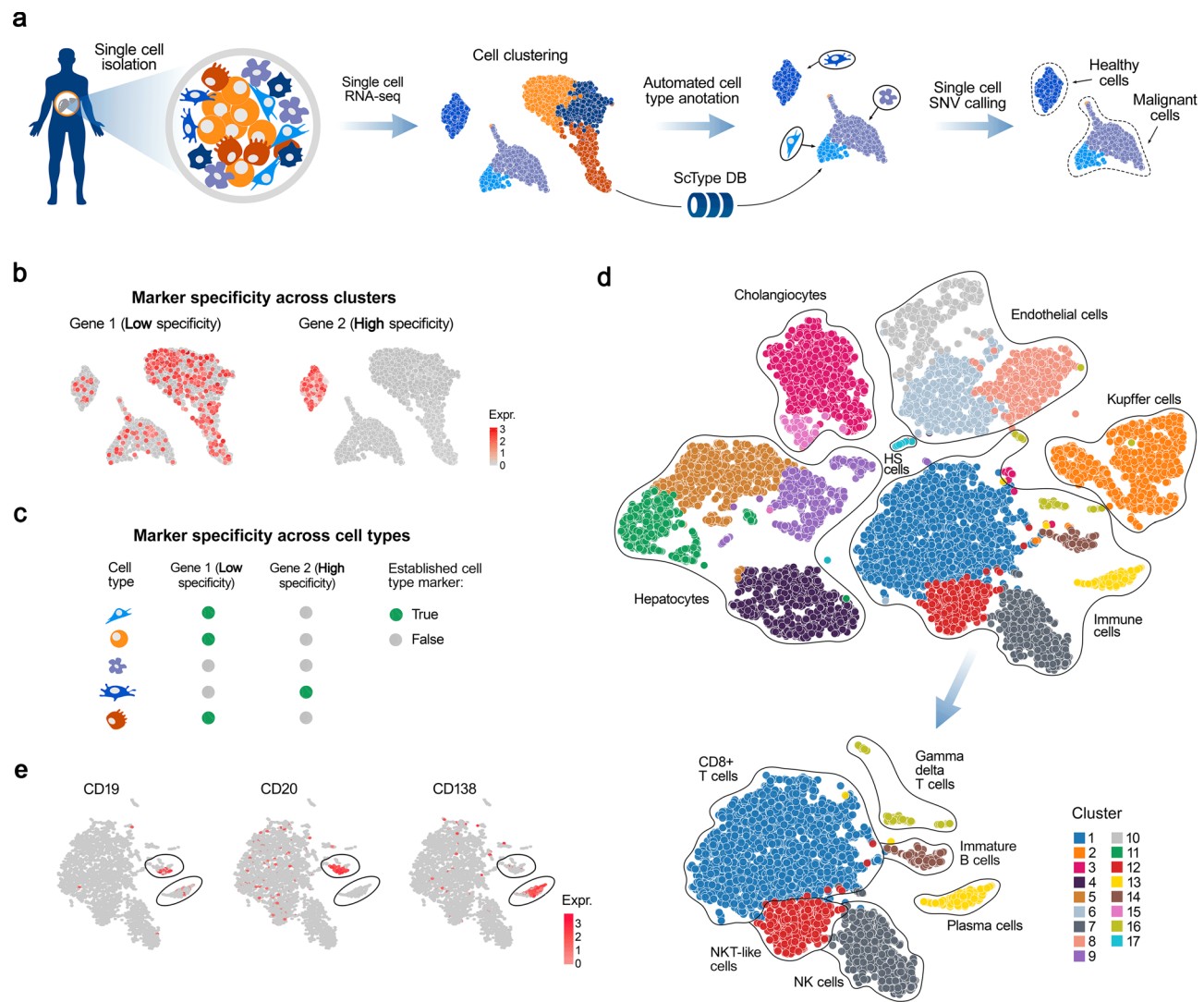

**Fig. 1 A schematic view of cell-type annotation using ScType. a** ScType requires only the raw or pre-processed single-cell transcriptomics dataset(s) as input. ScType implements options for additional quality control and normalization steps, where needed, followed by unsupervised clustering of cells based on scRNA-seq profiles. The results here are based on the Louvain clustering; however, also SC3, DBSCAN, GiniClust and k-means clustering options are available in ScType (see Methods). In the next step, ScType performs a fully-automated cell-type annotation using an in-built comprehensive marker database. Finally, ScType implements novel options for somatic single-cell SNV calling to distinguish between healthy and malignant cell populations. **b**, **c** ScType specificity score guarantees that the marker genes show specificity both across clusters and cell types for accurate unsupervised cell-type annotation with high cell subpopulation selectivity. **d** UMAP example of automated cell subtype identification by ScType in the liver atlas dataset, where it automatically labelled the same cell-types as assigned manually in the original study[10]. **e** Based on the information that plasma cells do not express common B-cell markers, such as *CD19* and *CD20*, but instead express CD138, ScType enhanced the resolution of cell-type annotations of two cell clusters, which were jointly annotated as B-cells in the original study, by segregating them into immature B-cell and plasma (B) cell types (lower UMAP plot of panel (**d**).

comparison to the cell-type annotations given by the original authors of additional four published scRNA-seq studies. We further utilized all the six datasets to compare ScType performance against the other recent cell-type annotation methods in terms of their accuracy and running time. The RNA-seq datasets used in the benchmarking originated from various tissues, including human liver[10], pancreas[24], peripheral blood mononuclear cells (PBMCs)[25], brain[26], as well as mouse lung[27] and retina samples[20]. These scRNA-seq datasets enabled us to investigate the performance of ScType and the related methods in the context of various sequencing platforms, tissues types and organisms.

Among the six scRNA-seq datasets from various human and mouse tissues, ScType correctly annotated a total of 72 cell types out of 73 cell-types (98.6% accuracy), including 8 correctly

reannotated cell-types that were originally incorrectly or non-specifically annotated (Fig. 2a). The only cell-type ScType was unable to automatically label as known was fetal cells in the human brain dataset, as there are no fetal cell markers available for human brain in the current version of the ScType database. However, ScType correctly identified all the other cell populations of the human brain tissues (oligodendrocytes, astrocytes, microglial cells, neurons, endothelial and oligodendrocyte precursor cells), according to the annotations made in the original study[26]. Furthermore, ScType was able to refine the originally-annotated neuron cell population into cholinergic (expressing *SLC17A7*)[28] and glutamatergic (expressing *ACHE*)[29] subtypes.

Next, we compared ScType against the state-of-the-art cell-type annotation methods with reported (i) highest accuracy (scSorter[30]

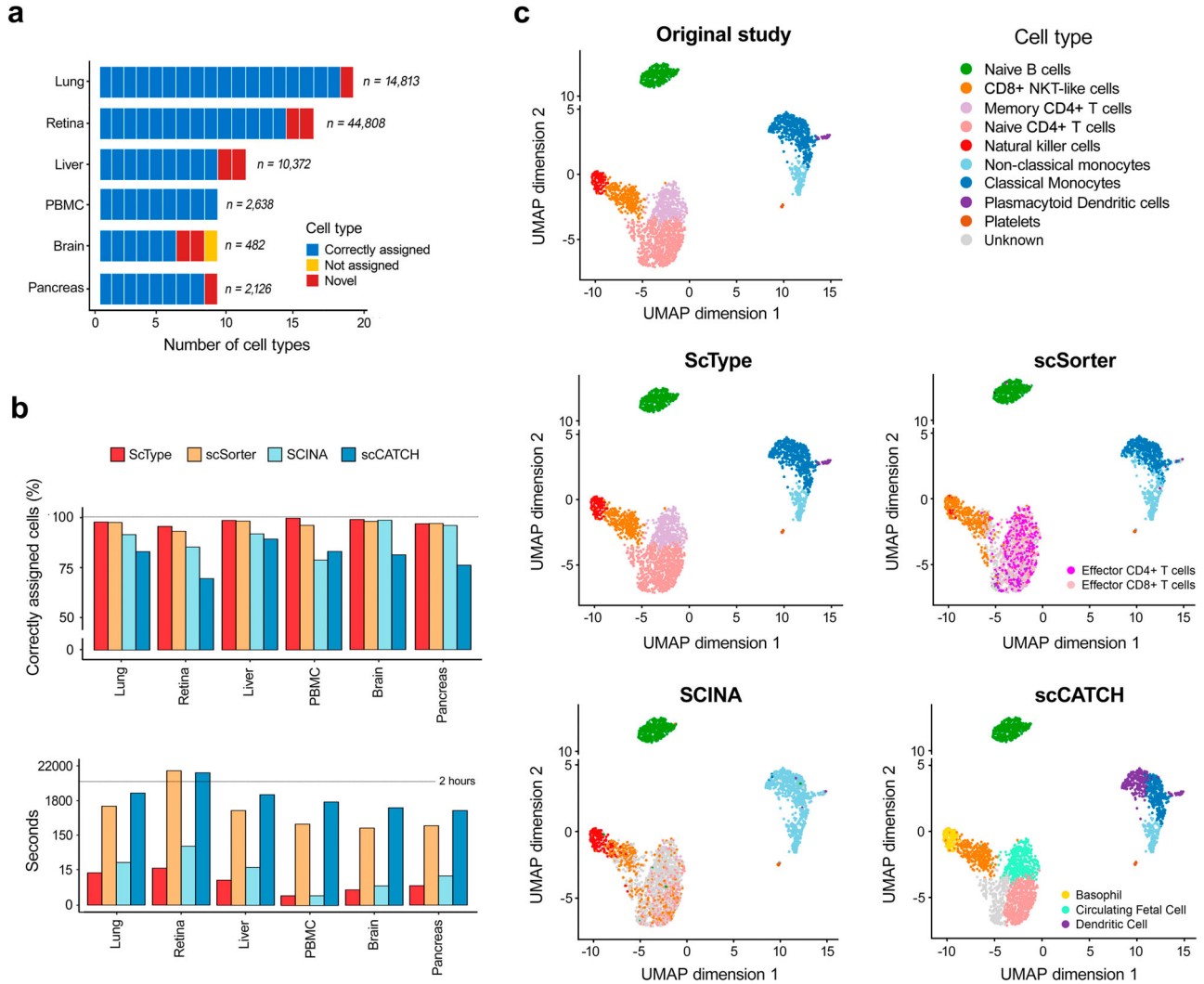

**Fig. 2 ScType performs ultra-fast and accurate cell-type annotation across various tissues. a** The overall performance of ScType across six human and mouse scRNA-seq datasets. ScType automatically assigned cell-types according to the original studies, and it also correctly reannotated five cell types in the human brain, liver and pancreas tissues, compared to the original studies (marked as novel cell types). ScType labelled only single cluster (fetal cells) as unknown cell-type in the human brain dataset (marked as not assigned). Similarly, in the mouse lung and retina datasets, ScType enabled automated identification of all the cell types, and it also correctly reassigned three novel mouse cell types. **b** Comparison of ScType with three recently developed cell-type annotation methods in terms of percentage of correctly annotated cells (upper panel) and running time (lower panel; *note*: the time axis is log-scaled). **c** Detailed cell-type annotations of the human PBMC dataset with the methods under comparison.

that was recently shown to outperform several popular tools such as Garnett[31] and CellAssign[32]), (ii) fastest running time (SCINA[33]), and (iii) full-automated process (scCATCH[34]). For an unbiased comparison, we provided scSorter and SCINA with our in-built database markers, while scCATCH utilized its own integrated marker database. Overall, ScType correctly annotated more than 94% of the cells in each dataset (Fig. 2b, upper panel), outperforming the other algorithms in 5 out 6 datasets. We note that the differences between ScType and the next best performing method scSorter were not large, as both methods showed a high accuracy in all the datasets, but importantly ScType was more than 30 times faster than scSorter (Fig. 2b, bottom panel). In particular, ScType showed almost perfect accuracy in the challenging human PBMC dataset (Fig. 2c), where there exist multiple closely-related subtypes. In contrast, scSorter and scCATCH did not identify natural killer cell populations, and they incorrectly identified several T cell subtypes, while SCINA was not able to distinguish between the two monocyte subpopulations as well as several subsets of T cells (Fig. 2c).

These benchmarking results indicate that ScType enables ultra-fast and highly accurate separation even between closely related subtypes by utilizing the novel concepts of marker gene specificity across both cell clusters and cell types.

**Evaluation of dropout effects and unknown cell types in simulated scRNA-seq data**. The dropout rates can be significant in single-cell data (up to 80%), which may notably impact the clustering solutions and annotation results. To investigate this effect, we utilized the Splatter method[35], and generated 45 simulated datasets with various dropout rates: 15 datasets with ~50% dropouts, 15 datasets with ~65% dropouts, and 15 datasets with ~80% dropouts (see Methods). For each cell type, we generated both highly-specific markers (ten top-ranked marker genes separately for each cell type), and low-specificity markers (ten top-ranked marker genes for a mixture of cell types including the cell type in question), which were used as the input for the ScType and the other methods (note: scCATCH was not used in this

comparison since its implementation does not allow for the use of custom markers). The comparison results in the simulated datasets showed that the ScType annotations are relatively robust against high percentages of cells with missing expression, with the annotation accuracy remaining above 90% even at 80% dropout rate, generally outperforming the other methods both in terms of the annotation accuracy and running time (Supplementary Fig. 2).

Due to the limited amount of information currently available on the markers expressed in many of the cell types, accurate identification of "unknown" cell types is an important practical task. To investigate the impact of unknown cell types on the performance of ScType, scSorter and SCINA methods, we utilized the same 45 simulated datasets and used a cross-validation scheme of "leave-one-cell-type-out" to evaluate the accuracy of identifying the unknown cell groups. More specifically, we removed the specific marker signatures of one of the cell types at a time in each of the 45 datasets, and then performed the cell type annotations with all the methods. In ScType, we considered a low ScType score (less than quarter the number of cells in a cluster) or a negative ScType score as low-confidence cell type annotations, which were assigned as "unknown" cell types. We observed that ScType and SCINA were able to correctly assign unknown cell types in most of the simulated datasets, 43 out of 45 datasets (95.5%) and 41 out of 45 datasets (91.1%), respectively, while ScSorter correctly identified unknown cell types only in 22 out of 45 datasets (48.8%).

**ScType utilizes both positive and negative markers for the cell type annotation**. As a unique feature, ScType accepts and makes use of not only positive markers, but also negative marker genes, i.e., markers that are not expected to be expressed in a particular cell type, with the aim to differentiate between closely related cell types. In general, we note that the same gene can be a positive marker for one cell type and a negative marker for another cell type. Supplementary Fig. 3 shows an illustrative example of how ScType calculates the marker specificity and enrichment scores based on both positive and negative marker genes. Compared to the number of positive markers, there is still only a relatively small number of negative markers in the current database, which originated from our literature search[36–40], but the users can apply their custom sets of negative (and positive) markers based on their domain knowledge and emerging studies to improve the annotation process and the coverage of the ScType marker database. This is expected to further increase the accuracy of automatic annotation of new scRNA-seq datasets in the future studies.

As an example, we found the negative markers to be informative when distinguishing between two closely-related groups of T-cells in the human PBMC dataset. It is known that both naïve and memory T cells express *CCR7* and *SELL* genes, which are required for lymph node migration, whereas these genes are not expressed in effector T cells[36,37]. Therefore, we added *CCR7* and *SELL* genes as negative markers for the effector T cells in the ScType database. When using both positive and negative markers, ScType was able to correctly distinguish naïve and memory T cells (Fig. 3a, left panel), whereas when using only positive markers, ScType assigned naïve T cells as effector T cells (Fig. 3a, right panel). In the latter case, an almost equal ScType score was assigned for both T cell types (4 and 7, respectively), even though *SELL* and CCR7 genes were expressed in this cluster (Fig. 3b). The original study also annotated the same cluster as naïve T cells[25], whereas the other methods were not able to correctly annotate the cluster (see Fig. 2c).

**Single-cell SNV calling distinguishes between healthy and malignant cell types**. To enable genetic analyses in cancer applications, we further implemented an option for single-

nucleotide variation (SNV) calling directly from the scRNA-seq data. As an example, we re-analyzed the scRNA-seq transcriptomic profile and cell-type composition of an AML patient sample from our recent study[18] using the ScType platform (Fig. 4a). After performing automated single-cell SNV calling (see Methods), we investigated whether the number of SNVs within a cell-type could distinguish between healthy and cancerous populations present in the patient sample (Fig. 4b). More specifically, ScType quantified the percentage of cells in a cell-type above the median SNV in the cancer consensus genes across all cells within the sample (ScType SNV score). As expected, we observed a higher SNV score in the *CD34+* progenitor (HSC/MPP) cells and *CD34+* interferon-stimulated gene (ISG)+ blast cells, as compared to *CD24+ CD66+* neutrophils and memory *CD+* T cells, which are usually considered as non-malignant cell types in AML[11] (Fig. 4c). As another validation for correctly distinguishing normal cells from malignant cells, we considered aneuploidy (unbalanced number of chromosomes), which is common for most human tumors[41]. To identify aneuploidy, we incorporated the recent Bayesian segmentation approach, CopyKAT[41], that classified majority of *CD2+ CD66+* neutrophils and memory *CD8+* T cells as diploid cells (Fig. 4d), suggesting their non-malignancy. These two validations demonstrated how ScType correctly assigned *CD2+ CD66+* neutrophils and memory *CD8+* T cells as non-malignant cells (ScType SNV score <20 and the majority of cell within the cell-type are classified as diploid cells, Fig. 4e).

To further investigate the ScType cell population classification, we studied the associations between the various cell-types based on the occurrence of common SNVs in the cancer consensus genes, and observed that non-malignant cell-types (i.e. memory *CD8+* T cells and *CD24+ CD66+* neutrophils) were closely similar to each other, while showing almost no SNV similarity with the malignant cell types (e.g. *HSC/MPP* and *ISG+* blast cells, see Fig. 4f). These results demonstrate how the ScType platform enables one to distinguish between malignant and non-malignant cell populations, based directly on scRNA data from a given patient sample, which is critical for the selection of safe and effective targeted treatment regimens for individual AML patients. For other cancer types and malignancies, the platform similarly supports automated options for the marker selection and cell population classification into healthy and diseased cells. In addition, ScType enables the visualization of the genome-wide copy number profiles from scRNA-seq data using CopyKAT to identify larger-scale copy number alterations (CNAs), such as somatic gains or deletions of large segments of chromosomes (see Supplementary Fig. 4 as an example in the AML patient sample). In the downstream analyses, the identified CNAs may explain difference in the cellular phenotypes of specific cell types and subclones, including their apoptotic potential or drug sensitivity. This case study shows how the general purpose ScType platform can be used for anticancer applications, where single-cell analyses are increasingly being used both for better understanding of the disease processes and precision treatment selection for individual patients.

**Discussion**
We presented ScType, a fully-automated platform for cell-type identification that enables accurate and ultra-fast single-cell-type annotations based solely on the given scRNA-seq data, using our comprehensive ScType marker databases as background information. To the best of our knowledge, ScType is currently the only unsupervised method that makes use of the marker gene specificity across both cell clusters and cell types to automatically identify highly-specific positive marker genes, along with negative

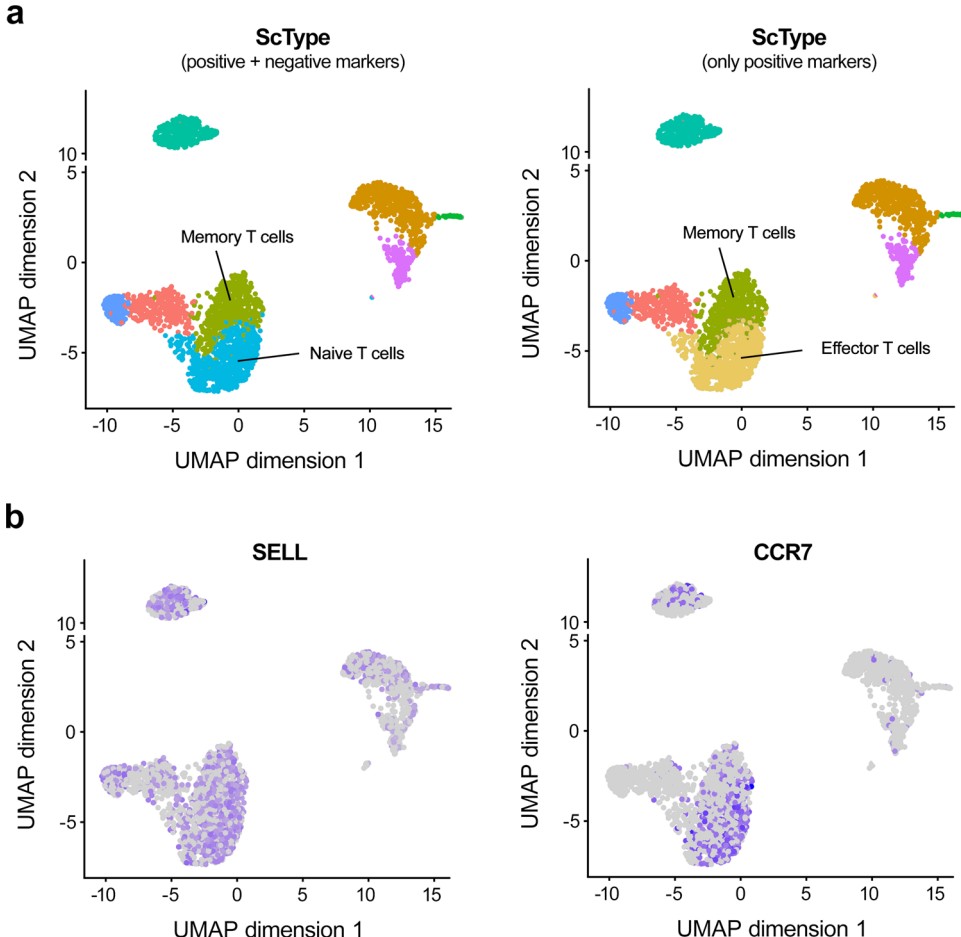

**Fig. 3 ScType utilizes negative markers when distinguishing between groups of T-cells. a** Automated cell type annotations with ScType in the human PBMC dataset when utilizing both positive and negative marker genes (left panel), and only positive marker genes (right panel). **b** Expression of *CCR7* and *SELL* genes, which are required for lymph node migration, and are therefore not expected to be expressed in the effector T cells (i.e., negative markers).

marker genes to provide evidence against cells being of a particular cell-type for cell-type selective annotations. To promote its wide application, either as a stand-alone tool or together with other popular single-cell data analysis software (e.g., Seurat[42], MAST[43], PAGODA[44]), we have deployed ScType both as an interactive web-platform (http://sctype.app), and as an open-source R implementation (https://github.com/IanevskiAleksandr/sc-type).

We anticipate the ScType platform will accelerate unbiased phenotypic profiling of cells when applied either to large-scale single-cell sequencing projects or smaller-scale profiling of patient-derived samples. For example, the integrative marker information in the ScType database may enable the identification of rare cell subtypes that have distinct combinations of molecular markers, suggesting specific functions and phenotypic profiles. We recently demonstrated how the ScType database provided information for patient-tailored identification of cancer-selective combinatorial therapies for relapsed AML patients, each with different genetic background and resistance mechanisms[18]. The ScType annotation algorithm, together with the novel methods to distinguish between malignant and non-malignant cells, are expected to enable design of targeted treatment regimens also for other cancer types.

The existing computational methods for automatic identification of cell types can be broadly categorized into two groups: (1) supervised methods that require carefully-annotated training datasets labelled with correct cell populations to train the classifiers (e.g. CaSTLe[45] and ACTINN[46] that annotate cell types based on pre-defined reference set of cells without the need of cell marker input), and (2) a prior knowledge-based methods that require either a marker gene set or a pre-trained classifier for the selected cell populations (e.g. scSorter[30], SCINA[33], and scCATCH[34]). The supervised methods may have severe limitations when annotating especially rare populations of cells, due to lack of reference data to train the machine learning algorithms. Furthermore, supervised methods are notoriously time-consuming to train, as well as error prone to technical artifacts in the training data, which affect their prediction ability for new scRNA-seq data[47].

Similarly, the prior knowledge-based cell classification approaches have certain limitations. For instance, their performance heavily depends on the available gene lists provided as markers for each cell type, typically obtained from manual literature search or matching to marker databases that are still suboptimal both in coverage and specificity. Ideally, one would like to use an appropriate number of specific markers to achieve a maximally accurate cell-type classification. However, most existing methods utilize a limited number of markers, thereby potentially masking the identification of a subpopulation of cells that do not express the selected marker genes. Furthermore, the use of inconsistent cell-type markers across experiments and laboratories may compromise the reproducibility of the findings[47]. These caveats become even more pronounced when the number of cell types and samples increases, thus preventing

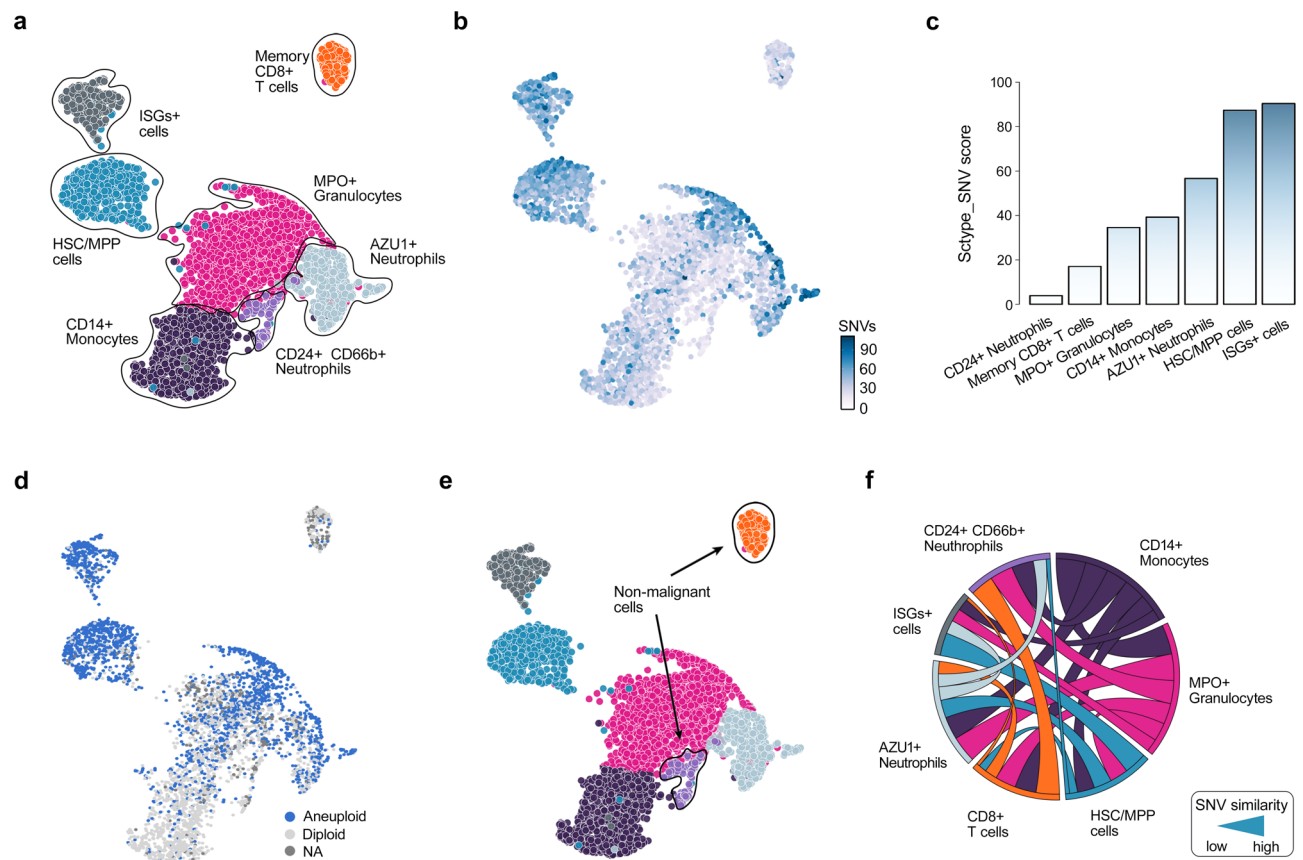

**Fig. 4 ScType enables SNV calling directly from the scRNA-seq profiles. a** UMAP plot of the cell types in the AML patient sample automatically annotated with ScType. **b** Distribution of somatic SNVs in the cancer consensus genes across the various cell types of the patient. **c** ScType SNV score summarizes the number of point mutations in the cancer genes for each cell type, shown as the percentage of SNVs above the median SNV value within the particular cell cluster. **d** UMAP showing aneuploid and diploid cell classification based on Bayesian segmentation approach CopyKAT[41]. **e** ScType assigns cell types as non-malignant when the ScType SNV score is below 20 and more than 50% of cells within the cell-type are classified as diploid. **f** Chord diagram shows the associations between different cell types in terms of the similarity of their SNVs in the cancer genes (i.e., occurrence of common SNVs, see Methods for details). The width of a connection corresponds to the degree of SNV similarity between cell-types, while the connection color indicates a specific cell-type as shown in the UMAP plot in panel a.

fast and reproducible annotations. It has been therefore argued that prior information does improve the automated cell-type identifications[32,47].

ScType implements a number of improvements compared to the existing cell-annotation tools. Our unsupervised approach outperformed the prior knowledge-based methods scSorter[30], SCINA[33], and scCATCH[34], which were recently shown to enable accurate annotation of multiple cell types[47]. Another group of supervised methods, such as CaSTLe[45], ACTINN[46], SingleR[48] and CHETAH[49], utilize reference bulk or single-cell transcriptomic data for cell-type predictions, and therefore require comprehensive, manually-annotated and high-quality reference datasets for training; furthermore, these methods do not allow identification of novel cell-type marker genes. In contrast, ScType requires neither reference scRNA-seq datasets nor manual selection of marker genes; instead, all the background information for established or de novo markers comes from the ScType database that is to date the most comprehensive database of specific markers for human and mouse cells.

In comparison with many other computational approaches that require manual interference[31,32], ScType takes a fully data-driven approach, and it annotates the cell-types at once in a totally unsupervised manner. The only input needed for the ScType tool is the raw sequencing data file, although uploading of pre-processed scRNA-seq data is also an option. This saves

considerable time and costs in the scRNA-seq analysis, especially when searching for cell-types in a tissue that involves a large variety of cell-types with similar transcriptomic profiles (e.g. bone marrow samples from mixed lineage leukemia subjects). The running time of ScType is also orders of magnitude faster than the supervised methods. Furthermore, ScType implements options for SNV and CNV calling from the raw scRNA-seq profiles of individual samples. The users may compare SNV levels across different cell types, and study associations between cell clusters based on their SNV load.

Using six scRNA-seq datasets from the human and mouse tissues, we demonstrated that ScType provides scalable and accurate identification of cell-clusters, and it is compatible with data formats from various sequencing techniques (e.g. Drop-seq and Smart-seq). These benchmarking results against the existing cell annotation approaches indicated that ScType is widely-applicable to various biomedical problems, and it provides fast and accurate cell-type classifications. Furthermore, we expect that the comprehensive ScType database will lead to the development of new and improved cell-type detection methods, as well as accelerate the implementation of single-cell pipelines for translational applications, such as monitoring of therapy resistant cancer cell sub-populations and designing of targeted combinatorial therapies to overcome the monotherapy resistance in cancer patients, which require fast and automated analyses for real-time clinical implementation.

Due to both biological and technical reasons, certain genes may not be detected in a specific cell type[50], which can be considered as a negative marker in ScType for the cell type. In the future extensions of ScType, we plan to incorporate correction methods for drop-out events in scRNA-seq data, such as scDoc[51], which should help to identify true and robust negative markers. Even if the number of negative markers is still relatively low, compared to the number of positive markers, we expect the annotation results of ScType will further improve once more negative markers are identified in future studies and incorporated into the ScType database. In addition, ScType utilizes z-score transformation for computationally-efficient combination of multiple markers, which may be suboptimal when dealing with single-cell transcriptomic data. However, our results in the simulated datasets showed a drastic decrease in the annotation accuracy when removing the z-score transformation ($P < 0.001$, Wilcoxon test; Supplementary Fig. 5).

In conclusion, ScType provides fully-automated cell-type identification using its own in-built marker database, as well as identification of malignant cell populations and cancer targets based on SNV calling and aneuploidy identification, requiring only raw scRNA-seq data as an input. As increased number of scRNA-seq datasets from various tissue types become available from the Human Cell Atlas and other projects, the accuracy and coverage of the ScType platform is expected to increase accordingly.

## Methods

**ScType database construction**. ScType database is the largest database to date of human and mouse cell-specific markers, compiled by integrating the information available in the CellMarker database (http://biocc.hrbmu.edu.cn/CellMarker/) and PanglaoDB (https://panglaodb.se), which are currently the two largest available databases for cell-type markers. In the CellMarker database, 13,605 cell markers for 467 cell types in 158 human tissues/sub-tissues and 9,148 cell makers for 389 cell types in 81 mouse tissues/sub-tissues were manually collected and curated from more than 100,000 published papers[8]. In the PanglaoDB, 6,631 gene markers mapping to 155 cell types have been identified by differential expression analysis in the particular cell types using single-cell data and a community-based crowdsourcing approach for curation of gene expression markers[9]. However, these two databases differ in the number of tissues, cell types and marker numbers, as well as in the way the markers have been assigned to each cell type. Therefore, we firstly converted the non-uniform gene IDs to approved gene symbols within and between the databases. Next, we removed the low evidence marker genes from the CellMarker database (i.e., genes having only one reference to support the cell-type marker), and genes that appeared in less than five clusters of specific cell-type from PanglaoDB. Additionally, we excluded genes showing no expression across all the datasets in PanglaoDB. Ultimately, we unified the cell and tissue naming from the two databases and excluded tissues comprising less than 5 cell types. Fifteen novel cell types with corresponding marker genes were added by manual curation of >10 papers to the current version of the compiled ScType database (https://sctype.app/database.php), as relatively few brain and eye tissue cell types were provided in the first version of the database. Furthermore, we extracted 37 negative markers for the cell types studied in this work based on literature search[36–40]. The users can also apply their custom sets of both positive and negative markers, based on their domain knowledge and emerging studies, both in R code and web-app, and the community can suggest new marker genes for the cell type annotation to be included into future versions of ScType database either using GitHub's "Pull Request" feature or by sending an email to the authors. In total, the current version of the ScType database comprises 3,980 cell markers for 194 cell types in 17 human tissues and 4,212 cell markers for 194 cell types in 17 mouse tissues.

**Cell-type specificity of markers**. Cell-type specificity score provides a quantitative measure of how uniquely a particular marker $i$ identifies a specific cell-type of the given tissue ($t$), with higher scores corresponding to highly-specific markers and lower scores to the promiscuous markers. The cell-type specificity score ($S$) was calculated separately for each marker gene $M_i$ within a tissue $t$ by firstly pooling all the cell-type specific markers within the tissue into marker pool $M$, and then calculating the cell-type specificity score for each marker as $S_i^t = 1 - \frac{|M_i|_t - min(|M|_t)}{max(|M|_t) - min(|M|_t)}$. Here, $|M_i|_t$ denotes the number of cell types of tissue $t$ where the $i$th marker is enlisted, $min(|M|_t)$ and $max(|M|_t)$ are the minimum and maximum number of cell types for which any of the provided genes is enlisted as a marker in the ScType database. A toy example of the calculation of the cell-type specificity score is shown in Supplementary Fig. 3a.

**ScType workflow options**. ScType provides a complete pipeline for single-cell RNA-seq data analysis and cell-type annotation. We utilized Seurat v4.0[42] for data processing and normalization. For clustering analysis, the default option is Louvain clustering based on a shared nearest neighbor graph (using FindClusters function with the resolution parameter set to 0.8 and 20 principal components given as input), which was used to generate the current results; however, also SC3[52], DBSCAN[53], GiniClust[54] and k-means clustering options are available in ScType. The clusters are visualized using either principal components analysis (PCA), t-distributed Stochastic Neighbor Embedding (t-SNE), Uniform Manifold Approximation and Projection for Dimension Reduction (UMAP), Isomap[55], Diffusion Map[56], largeVis[57] or by means of expression heatmaps. For the integrated multi-scRNA-seq dataset analysis, ScType uses FindIntegrationAnchors and IntegrateData functions from Seurat v4.0 that were shown to enable an effective identification of anchor correspondences across multiple single-cell datasets[42]. As a key unique component, ScType implements an ultra-fast and fully-automated cell-type identification, using highly-specific marker genes (see above), and allows the user to explore each gene's contribution to cell type annotations (see Supplementary Fig. 1). Finally, Sctype implements options for SNV calling and aneuploidy identification, and it automatically suggests separation between non-malignant and malignant cell types (see below).

**ScType cell-type annotation**. In order to assign each cell-type to a cluster ($p$), given the input scRNA-seq data ($X$) with $m$ genes and $n$ cells, ScType first standardizes each gene expression profile into z-score across all cells. Only positive and negative markers genes corresponding to different cell types of the specified tissues are considered (extracted either from ScType database or using user-provided custom cell-type gene sets as an alternative option). Then, each gene expression level is multiplied with its cell type-specificity ($S_i^t$) score: $X' = ((Z(X^T))^T \subseteq M_t) \cdot S_i^t$, where $X'$ is a transformed scRNA-seq expression matrix of $n$ cells and $|M_t|$ genes, $M_t$ is the vector of marker genes of all cell types within the tissue $t$, and $Z$ denotes the z-score-transformation. The transformed expression values for each cell-type ($c$) are further summarized into cell type-specific marker-enrichment-score as the normalized sum of all the individual genes supporting a cell-type:
$x'_c = \frac{\sum_{i=1}^j x'_i}{\sqrt{j}} - \frac{\sum_{k=1}^l x'_k}{\sqrt{l}}$, where $x'$ is the unique column of $X'$ corresponding to one cell, $c$ is the specific cell-type within the tissue, and $i, \ldots, j$ are the indices corresponding to cell-type-specific marker genes, while $k, \ldots, l$ are the indices of negative marker genes that are not expected to be expressed in the cell type. Such transformation results in normalized expression matrix of c-by-n dimension, where each row represents one of the cell types and each column represents an individual cell. Finally, by summing up the values of each row (cell type) across the cells corresponding to a specific cluster $p$, the cluster summary enrichment-score (called ScType score, Supplementary Fig. 3b) for each cell-type is calculated:
$ScType\ score_c = \sum_{z \in p} x_c^z$. A cell-type with the highest ScType score is used for assignment to the cluster $p$. Such formulation guarantees marker gene specificity across both the cell-types ($S_i^t$) and cell clusters ($\sum_{z \in p} x_c^z$) (Fig. 1b, c), thus allowing for a highly accurate cell-type annotation. We consider a low ScType score (less than quarter the number of cells in a cluster), or a negative ScType score to indicate a low-confidence cell-type annotation, which are assigned as "unknown" cell type in the web-tool. In addition to the cell-type assignments, the ScType web-portal (http://sctype.app) allows users to view the metadata based on which the assignment was made, view the markers that are enriched in each specific cluster, and plot the cumulative gene-specificity for different cell types as bar graphs.

**Systematic comparison in publicly available datasets**. In order to benchmark the ScType against the other cell-type annotation methods, we utilized six scRNA-seq datasets from public domain and re-analysed these using ScType platform. Five datasets were downloaded from Gene Express Omnibus (GEO): Human Liver (GSE124395), Human Brain (GSE67835), Human Pancreas (GSE85241), Mouse Lung (GSE63269) and Mouse Retina (GSE63473). Human PBMC dataset was downloaded from the 10x Genomics Dataset Repository (https://s3-us-west-2.amazonaws.com/10x.files/samples/cell/pbmc3k/pbmc3k_filtered_gene_bc_matrices.tar.gz).

The human brain dataset was used to define the markers in the ScType marker gene database based on the publication list in the PangloDB paper[9] and website (https://panglaodb.se/papers.html). To make the comparison unbiased, we excluded the markers identified from the human brain data while annotating the cell cluster in the comparative study. The other marker source, CellMarker database, is based on manual curation of more than 100 000 publications[8], but it did not list the studies used for defining its marker genes. Hence, we could not confirm whether the CellMarker database have incorporated markers from the experimental dataset used in the comparative study.

**Comparison with other cell-type annotation tools**. We compared the accuracy, running time and requirement of hardware resources of ScType against scSorter[30], SCINA[33], and scCATCH[34] using the six scRNA-seq datasets (see Publicly available datasets). We used the default parameters to run all the methods. To provide unbiased comparisons, we used the same cell-type marker gene information (based on our ScType database) in scSorter and SCINA. scCATCH has its own in-built database that was used in the analysis.

We utilized the Splatter method[35] to generate 45 simulated datasets with various dropout rates: 15 datasets with ~50% dropouts, 15 datasets with ~65% dropouts, and 15 datasets with ~80% dropouts (dropout.shape -0.5 and dropout.mid −2.5, 0, 2.5, respectively). Each simulated dataset contained 10 cell types with varying cell proportions using scaled lognormal random deviates (e.g., 42.3%, 22.7%, 12.6%, 4.5%, 4.1%, 4.0%, 2.9%, 1.8%, 0.3%). Each dataset has "Moderate" similarity among the cell types (i.e., de.facScale parameter was randomly set to either 0.5, 0.6 or 0.7). Using the simulated datasets, we also compared the performance and running time of ScType, scSorter, and SCINA methods. We note that the scCATCH method was not used in these comparisons since its implementation does not allow for the use custom markers.

We extracted marker genes from the simulated reference data by performing differential expression analysis using the Wilcoxon rank sum test, similarly to the recent comparative evaluation work[58]. For each cell type, we generated both highly-specific markers (ten top-ranked marker genes separately for each cell type), identified through differential expression, and low-specificity markers (ten top-ranked marker genes for a mixture of all simulated cell types including the cell type in question), which were used as the input for the ScType and other methods. To generate low-specificity markers for $n^{\text{th}}$ cell type, we took all the pairs of two cell types including nth cell type (e.g. nth and $n+1$, or nth and $n+2$ cell types), and identified differentially expressed genes between each group of two cell types vs other cells; finally, we combined all "unspecific" genes together and randomly selected 10 of them.

**SNV identification using single-cell RNA sequencing**. ScType utilizes raw scRNA-seq data to identify single-nucleotide variants (SNVs) present in each cell. ScType processes the raw input scRNA-seq BAM file using samtools[59] and call the SNVs using Strelka2[60]. Next, ScType connects the SNV to each cluster using vartrix (https://github.com/10XGenomics/vartrix). As an extended feature, ScType also calculates the sum of total number of SNV present in the COSMIC cancer census genes[61] as the combined SNV score (ScType SNV score) for each cluster in a cancerous tissue profile. More specifically, ScType SNV score summarizes the number of point mutations in the cancer genes for each cell type, calculated as the percentage of SNVs above the median SNV value within the particular cell cluster.

ScType also incorporates the recently implemented Bayesian segmentation approach, called CopyKAT[41], to distinguish between aneuploid and diploid cells. Based on these two analyses, ScType automatically makes a classification between malignant and non-malignant cells. ScType assigns cells as non-malignant if ScType SNV score <20 and the majority of cell within the cell-type (>50%) are classified as diploid and the others as malignant. Users can use these two analyses to assign healthy and cancerous cell-type labels to the cell clusters, based on the assumption that cancerous cell clusters tend to have more SNVs in the cancer genes and aneuploidy is common for most human tumors[41]. The list of cancer consensus genes was downloaded from the catalogue of somatic mutation in cancer database (COSMIC v92)[61].

**Reporting summary**. Further information on research design is available in the Nature Research Reporting Summary linked to this article.

## Data availability
Publicly available scRNA-seq datasets for methods benchmarking: Human Liver (GSE124395) Human Brain (GSE67835), Human Pancreas (GSE85241), Mouse Lung (GSE63269), and Mouse Retina (GSE63473); Human PBMC dataset was downloaded from the 10x Genomics Dataset Repository [https://s3-us-west-2.amazonaws.com/10x.files/samples/cell/pbmc3k/pbmc3k_filtered_gene_bc_matrices.tar.gz].
Previously published scRNA-seq data from an AML patient sample[18]: EGAS00001004614. Source data are provided with this paper.

## Code availability
R source-code of the ScType algorithm is available at https://github.com/IanevskiAleksandr/sc-type/ to allow reproduction of the results and its further comparison against or integration with other algorithms[62]. ScType is also freely available as an interactive web-tool at http://sctype.app. The ScType database is freely available at https://sctype.app/database.php. Quick start instructions and a step-by-step analysis example is provided at GitHub (https://github.com/IanevskiAleksandr/sc-type/), and a step-by-step tour that guides the new users through the web-application, as well as documentation of the analysis steps is provided on the landing-page of the app (http://sctype.app).

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

## Acknowledgements

The authors thank Dr. Pirkko M Mattila, Dr. Jenni Lahtela and Bhiswa Ghimire for their valuable suggestions on how to improve the web-tool, Olle Hansson for the FIMM cluster server machine to host the web-tool and the database, and all the beta-testers for confirming the smooth operation and reproducibility of the analyses. This work was supported by the Academy of Finland (grants 295504, 310507, 326238, 340141 and 344698 to TA), European Union's Horizon 2020 Research and Innovation Programme (ERA PerMed JAKSTAT-TARGET), the Cancer Society of Finland (TA), the Sigrid Jusélius Foundation (TA), and the Norwegian Cancer Society (grant 216104 to TA).

## Author contributions

A.I., A.K.G. and T.A. conceived and planned the study. A.I. developed the method, implemented the ScType web-tool and collected and analyzed the data. A.I. compiled the ScType database with the help of A.K.G. A.I. prepared the figures for manuscript with the help of T.A. and A.K.G. T.A. and A.K.G. supervised the study. All the authors wrote the manuscript and approved its final version.

## Competing interests

The authors declare no competing interests.
