## [Peer Review File · Nature Communications]

Fully-automated and ultra-fast cell-type identification using specific marker combinations from single-cell transcriptomic dataREVIEWER COMMENTS

Reviewer #1 (Expertise: single cell annotation methods):

Review of NCOMMS-21-22446

Fully-automated and ultra-fast cell-type identification using specific marker combinations from single-cell transcriptomic data

This manuscript describes *ScType*, an automated method for cell-type annotation of single cell RNA-seq dataset. It is well written and straightforward to follow. The authors demonstrated that sc-type was the most accurate and fastest unsupervised cell-type identification methods. It appears the cell-type annotation algorithm itself is not so novel but their strength lies in the use of their own marker database that is an integration of CellMarker and PanglaoDB. Another interesting aspect of *ScType* is the use of negative markers that should be expressed in a given cell-type. Close examination of the marker database available in their web portal did not manifest any negative markers. It appears the ScType app does not have an input mechanism to accept negative markers. It would be desirable to exemplify the advantage of using negative markers. The GUI in the web portal looks nice and easy to use. I have a few technical comments as follows:

I cannot follow the paragraph shown below:

489 **Cell-type specificity of markers**

490

491 Cell-type specificity (S) was calculated separately for each marker gene (M_i) across the cell types,
492 hence providing a quantitative measure of how frequently the marker maps to the cell-type
493 uniquely within a particular tissue (t) using the cell-type specificity score: $S_i^t =$
494 $\frac{|M_i|_t - \min(|M|_t)}{\max(|M|_t) - \min(|M|_t)}$, where $M = (M_1, \dots, M_i, \dots, M_m)$, and m is the total number of marker genes M
495 present across all the cell types of the tissue t , and i is the index of each unique marker.

Apparently, the cell-type specificity score should be calculated for each cell-type of the given tissue t and for each marker gene i . As the equation shown in line 494 has no provision for different cell types, how can it produce cell-type specificity score? What does $|M_i|_t$ mean? Does it mean the number of the cell-types of tissue t where the i -th marker is enlisted? Maybe the authors want to produce one score for each marker for a given tissue. The authors may have intended to emphasize how unique to a specific cell-type is a marker. If my understanding of $|M_i|_t$ is right, the equation in line 494 would be smaller for specific markers, while it becomes larger for promiscuous markers. This might be opposite to the authors' intention.

In the following paragraph, z-transformation of the expression level is mentioned.

515 **ScType cell-type annotation**

516 In order to assign each cell-type to a cluster (p), given the input scRNA-seq data (X) with m genes
517 and n cells, ScType first standardizes each gene expression profile into z-score across all **genes**.

Should each gene be z-transformed across all cells, not genes?

The following paragraph defines cell type-specific marker-enrichment score.

524 each cell-type (c) are further summarized into cell type-specific marker-enrichment-score as the
525 normalized sum of all the individual genes supporting a cell-type: $x'_c = \frac{\sum_{i=1}^j x'_i}{\sqrt{j}} + \frac{\sum_{k=1}^l x'_k}{\sqrt{l}}$, where x' is
526 the unique column of X' corresponding to one cell, c is the specific cell-type within the tissue, and
527 i, \dots, j are the indices corresponding to cell-type-specific marker genes, while k, \dots, l are the indices of
528 negative marker genes that are not expected to be expressed in the cell type. Such transformation

The genes k, \dots, l are negative markers. Shouldn't their expression values be subtracted in line 525?

I have tested the web portal at <http://sctype.app> using the one of the example datasets supplied at the site. Namely, the 10X Genomics dataset processed by the cellranger pipeline was downloaded from the web site and uploaded back to the server. I have started the analysis using the suggested default values. The QC step failed as shown below:

I was not able to verify the integrity of the web portal. Moreover, the authors claim the R source-code is available from the URL given in line 578 as shown below:

576 **Code and data availability**

577 The R source-code of the ScType algorithm is freely available at
578 https://sctype.app/source_code.php to allow reproduction of the results and its further comparison
579 against or integration with other algorithms. ScType is also freely available as an interactive web-
580 tool at <http://sctype.app>. The ScType database is freely available at
581 <https://sctype.app/database.php>.

The URL redirects to <https://github.com/lanevskiAleksandr/sc-type/>. I could not find any R codes there.

Reviewer #2 (Expertise: scRNASeq for the analysis haematopoetic datasets):

The manuscript by lanevsky et al. describes ScType, a new platform for the analysis of single-cell RNAseq data sets available as an interactive web tool and R source code. Its main features are a novel tool for the unsupervised, marker-based annotation of the identified cell clusters and the implementation of specific methods for analyzing cancer-related single-cell experiments.

Annotation of cell clusters is performed through a scoring function in which normalized marker-genes expression values are multiplied by a novel, tissue-specific, marker-gene cell-specificity index. ScType uses a marker-genes database obtained by combining 2 of the most significant resources currently available in the literature (CellMarker and PanglaoDB) with the addition of cell-type markers derived from the manual curation of recent publications aimed at covering tissues underrepresented in the original version of the database. Clusters annotation is a common and essential step of single-cell transcriptomic data analysis, and several methods have been proposed to perform this task. ScType outperformed most alternative approaches in annotating clusters with the appropriate cell-type labels in a comparative study based on published experimental data sets. In cases where ScType cannot provide a significant gain in accuracy, it gives the results in a considerably shorter amount of time.

In single-cell RNA data sets involving cancer-related samples, ScType offers the possibility to identify which clusters are enriched in malignant/healthy cells by simultaneously analyzing the ScType_SNV score distribution and the results generated by CopyKAT. ScType_SNV score is an index that measures how mutations in the cancer-consensus-gene list are distributed among the cells and clusters. CopyKAT instead classifies as aneuploid (malignant) or diploid (healthy) individual cells using a Bayesian segmentation approach. As an example of analysis performed on a cancer data set, authors re-analyses the data from one of their previous publications regarding a sample isolated from an AML patient.

1. Contrary to many alternative methods, ScType takes advantage of both positive and negative markers for annotating cell clusters. However, it is known that, for both biological and technical reasons, many genes considered as markers are in reality detected (non-zero) only in a fraction of the cells genuinely belonging to a specific cell type [Guo, H., & Li, J. (2021)]. Since the percentage of cells with missing expression can be significant (up to 80%), this can significantly impact smaller clusters.

2. Despite being a standard step in many single-cell RNA analysis workflows, the z-score transformation is suboptimal when dealing with single-cell transcriptomic data. I am aware that this step might be difficult to replace with alternative procedures, but it represents a limitation worth mentioning if not throughoutly investigated.

3. In the manuscript's main text, it is mentioned that ScType label as “unknown” a cluster corresponding to a cell type not included in the reference database. Still, it is not explained which criteria have been used for this “missing” annotation. Given the limited amount of information still available nowadays about the markers expressed by many cell types, correctly identifying “unknown” cell types is crucial and has to be stated clearly how this is done.

To address issues 1,2 and 3 aforementioned, I would suggest running a comparative study based on data generated with a method designed to mimic single cell transcriptional profiles [Zappia, L et al. (2017); Zhang, X., Xu, C., & Yosef, N. (2019), Dibaenia, P., & Sinha, S. (2020)]. The data set could be composed of clusters corresponding to a variable number of cell types with known and unknown marker genes. This would allow to:

1. Compare ScType to alternative methods under controlled scenarios.
2. Investigate the impact of negative markers on minor cluster annotation.
3. Measure how z-score transformation affects cell type identification.
4. Test the performance of ScType in detecting “unknown” cell types.

4. The ScType web tool is well designed, intuitive, and responsive. The platform runs smoothly

using the default data set provided by the authors as examples. When I tried to analyze my data set, I uploaded the data (count matrix of 8K human cells) and performed the 2 quality control steps (filtering of cells and selecting variable genes). However, the remaining and most relevant parts of the workflow did not work. I tried multiple operating systems and browsers with no success.

Minor comments

1. In general, I felt the first part of the manuscript focused on the unsupervised cluster annotation and the second, more cancer-oriented, a bit unrelated. Both are interesting and promising, but I think they might have a slightly different audience.
2. A simple toy example showing how the cell-type specificity-score and the enrichment-score are calculated on a small set of marker genes would make it easier for readers to have an intuition of how ScType annotation works.
3. It is not clear whether the experimental data sets used in the comparative study have also been used to define cell type marker-gene included in the database. Clarify this point since it is relevant in the evaluation of the performance of ScType.

REVIEWER COMMENTS

Reviewer #1 (Expertise: single cell annotation methods):

This manuscript describes ScType, an automated method for cell-type annotation of single cell RNA-seq dataset. It is well written and straightforward to follow. The authors demonstrated that sc-type was the most accurate and fastest unsupervised cell-type identification method. It appears the cell-type annotation algorithm itself is not so novel but their strength lies in the use of their own marker database that is an integration of CellMarker and PanglaoDB. Another interesting aspect of ScType is the use of negative markers that should be expressed in a given cell-type.

We thank the reviewer for positive comments and appreciating our work toward addressing the challenge of developing a fully-automated and ultra-fast method for cell type annotation of large-scale scRNA-seq data. We are also extremely grateful for pointing out the below points for improvement of the web-portal and the mistakes in the mathematical equations, which we have now been corrected (please see below)

Close examination of the marker database available in their web portal did not manifest any negative markers. It appears the ScType app does not have an input mechanism to accept negative markers. It would be desirable to exemplify the advantage of using negative markers.

We thank the reviewer for the excellent comment. We have now clearly marked the negative marker genes used in the current ScType database in the web-portal (the last columns on the right hand-side of the marker table): <https://sctype.app/database.php>.

As expected, the number of negative markers is still relatively low, compared to the positive markers, but we expect to see further improvements in the annotation results of ScType once more negative markers are identified in the future studies and used in ScType.

The option for adding new negative markers is now also clearly described in the step-by-step example at <https://github.com/lanevskiAleksandr/sc-type#negativemarkers>, and the use of negative markers is further described as one of (optional) input parameters for `sctype_score` function: https://github.com/lanevskiAleksandr/sc-type/blob/master/R/sctype_score.R/

In the web-app, the users can apply their custom sets of both positive and negative markers, based on their domain knowledge and emerging studies, and the community can suggest new marker genes for the cell type annotation to be included into future versions of ScType database either using GitHub's "Pull Request" feature or by sending an email to the authors.

As an example, we have now demonstrated how to utilize negative markers to differentiate between two close (in terms of differentiation) subsets of T-cells in human PBMC dataset, and added the below paragraph and figure to the revised manuscript (pages 8-9):

ScType utilizes both positive and negative markers for the cell type annotation

As a unique feature, ScType accepts and makes use of not only positive markers, but also negative marker genes, i.e., markers that are not expected to be expressed in a particular cell type, with the aim to differentiate between closely related cell types. In general, we note that the same gene can be a positive marker for one cell type and a negative marker for another cell type. Fig. S3 shows an illustrative example of how ScType calculates the marker specificity

and enrichment scores based on both positive and negative marker genes. Compared to the number of positive markers, there is still only a relatively small number of negative markers in the current database, which originated from our literature search³⁶⁻⁴⁰, but the users can apply their custom set of negative (and positive) markers based on their domain knowledge and emerging studies to improve the annotation process and the coverage of the ScType marker database. This is expected to further increase the accuracy of automatic annotation of new scRNA-seq datasets in the future studies.

As an example, we found the negative markers to be informative when distinguishing between two closely-related groups of T-cells in the human PBMC dataset. It is known that both naïve and memory T cells express CCR7 and SELL genes, which are required for lymph node migration, whereas these genes are not expressed in effector T cells^{36,37}. Therefore, we added CCR7 and SELL genes as negative markers for the effector T cells in the ScType database. When using both positive and negative markers, ScType was able to correctly distinguish naïve and memory T cells (Fig. 3a, left panel), whereas when using only positive markers, ScType assigned naïve T cells as effector T cells (Fig. 3a, right panel). In the latter case, an almost equal ScType score was assigned for both T cell types (4 and 7, respectively), even though SELL and CCR7 genes were expressed in this cluster (Fig. 3b). The original study also annotated the same cluster as naïve T cells²⁵, whereas the other methods were not able to correctly annotate the cluster (see Fig. 2c).

Figure 3. ScType utilizes negative markers when distinguishing between groups of T-cells. (a) Automated cell type annotations with ScType in the human PBMC dataset when utilizing both positive and negative marker genes (left panel), and only positive marker genes (right panel). **(b)** Expression of CCR7 and SELL genes, which are required for lymph node migration, and are therefore not expected to be expressed in the effector T cells (i.e., negative markers).

We would like to also note that in some cases, due to both biological and technical reasons, a certain gene may not be detected in a specific cell type, which would be considered as a negative marker in ScType. To account for this, we have added the below paragraph to the discussion section (page 13):

Due to both biological and technical reasons, certain genes may not be detected in a specific cell type⁵⁰, which can be considered as a negative marker in ScType for the cell type. In the future extensions of ScType, we plan to incorporate correction methods for drop-out events in scRNA-seq data, such as scDoc⁵¹, which should help to identify true and robust negative markers. Even if the number of negative markers is still relatively low, compared to the number of positive markers, we expect the annotation results of ScType will further improve once more negative markers are identified in future studies and incorporated into the ScType database.

50. Skinnider M, et al. Cell type prioritization in single-cell data. Nat Biotechnology 39, 30-34 (2021).

51. Ran, Di, et al. scDoc: correcting drop-out events in single-cell RNA-seq data. Bioinformatics 36, 4233-4239 (2020).

The GUI in the web portal looks nice and easy to use.

We thank the reviewer for appreciating our web-portal.

I have a few technical comments as follows:

I cannot follow the paragraph shown below:

489 **Cell-type specificity of markers**

490

491 Cell-type specificity (S) was calculated separately for each marker gene (M_i) across the cell types,

492 hence providing a quantitative measure of how frequently the marker maps to the cell-type

493 uniquely within a particular tissue (t) using the cell-type specificity score: $S_i^t =$

494 $\frac{|M_i|_t - \min(|M|_t)}{\max(|M|_t) - \min(|M|_t)}$, where $M = (M_1, \dots, M_i, \dots, M_m)$, and m is the total number of marker genes M

495 present across all the cell types of the tissue t , and i is the index of each unique marker.

Apparently, the cell-type specificity score should be calculated for each cell-type of the given tissue t and for each marker gene i . As the equation shown in line 494 has no provision for different cell types, how can it produce cell-type specificity score? What does $|M_i|_t$ mean? Does it mean the number of the cell-types of tissue t where the i -th marker is enlisted? Maybe the authors want to produce one score for each marker for a given tissue. The authors may have intended to emphasize how unique to a specific cell-type is a marker. If my understanding of $|M_i|_t$ is right, the equation in line 494 would be smaller for specific markers, while it becomes larger for promiscuous markers. This might be opposite to the authors' intention.

We thank the referee for pointing out this discrepancy and apologize for the sub-optimal explanation in the text that led to such confusion. Please find below answers to the specific questions:

As the equation shown in line 494 has no provision for different cell types, how can it produce cell-type specificity score?

A cell-type specificity score (S_i^t) for a marker i in a tissue t is calculated for each marker gene across all cell types of the given tissue (i.e. a single score for the particular gene i within the tissue t), with the aim to penalize for promiscuous markers expressed in multiple cell types. In other words, the cell-type specificity score prioritizes specific markers (uniquely expressed in a certain cell type) and de-prioritizes promiscuous markers (expressed in several cell types). Therefore, the correct equation for the cell-type specificity score is actually $S_i^t = 1 - \frac{|M_i|_t - \min(|M|_t)}{\max(|M|_t) - \min(|M|_t)}$, which corresponds to the scales::rescale R function that was used in the R-code and in the web-portal. We apologize for missing the 1- part in the manuscript text, which has now been corrected in the revised version (pages 17-18).

What does $|M_i|_t$ mean? Does it mean the number of the cell-types of tissue t where the i -th marker is enlisted?

Yes, $|M_i|_t$ denotes the number of cell types of tissue t where the i -th marker is enlisted.

We have now rewritten the corresponding paragraph with the corrected equation and an improved explanation (pages 17-18):

Cell-type specificity of markers

Cell-type specificity score provides a quantitative measure of how uniquely a particular marker i identifies a specific cell-type of the given tissue (t), with higher scores corresponding to highly-specific markers and lower scores to the promiscuous markers. The cell-type specificity score (S) was calculated separately for each marker gene M_i within a tissue t by firstly pooling all the cell-type specific markers within the tissue into marker pool M , and then calculating the cell-type specificity score for each marker as $S_i^t = 1 - \frac{|M_i|_t - \min(|M|_t)}{\max(|M|_t) - \min(|M|_t)}$. Here, $|M_i|_t$ denotes the number of cell types of tissue t where the i -th marker is enlisted, $\min(|M|_t)$ and $\max(|M|_t)$ are the minimum and maximum number of cell types for which any of the provided genes is enlisted as a marker in the ScType database. A toy example of the calculation of the cell-type specificity score is shown in Fig. S3a.

In the following paragraph, z-transformation of the expression level is mentioned.

515 **ScType cell-type annotation**

516 In order to assign each cell-type to a cluster (p), given the input scRNA-seq data (X) with m genes

517 and n cells, ScType first standardizes each gene expression profile into z-score across all genes.

Should each gene be z-transformed across all cells, not genes?

We thank the referee for pointing out this mistake. Yes, it should be “across all cells”. We have now corrected the manuscript text accordingly (page 18).

The following paragraph defines cell type-specific marker-enrichment score.

524 each cell-type (c) are further summarized into cell type-specific marker-enrichment-score as the
525 normalized sum of all the individual genes supporting a cell-type: $x'_c = \frac{\sum_{i=1}^j x'_i}{\sqrt{j}} + \frac{\sum_{k=1}^l x'_k}{\sqrt{l}}$, where x' is
526 the unique column of X' corresponding to one cell, c is the specific cell-type within the tissue, and
527 i, \dots, j are the indices corresponding to cell-type-specific marker genes, while k, \dots, l are the indices of
528 negative marker genes that are not expected to be expressed in the cell type. Such transformation

The genes k, \dots, l are negative markers. Shouldn't their expression values be subtracted in line 525?

Yes, the expression values of the negative markers are indeed subtracted, since we multiply them by -1 in the code and then take the sum, which is basically the same thing. We have now modified the manuscript text accordingly (page 18).

I have tested the web portal at <http://sctype.app> using the one of the example datasets supplied at the site. Namely, the 10X Genomics dataset processed by the cellranger pipeline was downloaded from the web site and uploaded back to the server. I have started the analysis using the suggested default values. The QC step failed as shown below:

I was not able to verify the integrity of the web portal.

We deeply apologize for the inconvenience caused. There was indeed a problem caused by the automated update of the PHP version, that we noticed after the submission. We have now disabled all the automated updates, ensuring this problem will not appear anymore. We have now carefully tested that everything is fixed and working properly. The Reviewer

2 also reported that the platform runs smoothly using the default data set provided by the authors as examples (see Reviewer 2, comment 4, below). We have also asked our colleagues in various sites to test the proper operation of the web-portal and the R-code and to reproduce the same results based on the step-by-step tour that we have now provided for the web-tool usage. In total, there have been so far 5 in-house beta-users and 4 outside users of ScType, and they have been able to analyze their own datasets, indicating robust functioning of the webtool.

Moreover, the authors claim the R source-code is available from the URL given in line 578 as shown below:

576 **Code and data availability**

577 The R source-code of the ScType algorithm is freely available at
578 https://sctype.app/source_code.php to allow reproduction of the results and its further comparison
579 against or integration with other algorithms. ScType is also freely available as an interactive web-
580 tool at <http://sctype.app>. The ScType database is freely available at
581 <https://sctype.app/database.php>.

The URL redirects to <https://github.com/lanevskiAleksandr/sc-type/>. I could not find any R codes there.

We have now made the R-codes available on the site, as well as added Quick start instructions and a step-by-step analysis example to demonstrate how to perform cell type annotation in an example data (<https://github.com/lanevskiAleksandr/sc-type/>). For non-computational users, we have also provided a documentation and step-by-step tour on the landing-page of the website (<http://sctype.app>) that guides the new users through the web-application, its input data formats as well as the QC and the analysis steps. These instruction documents are also uploaded as Supplementary files in the submission.

At the end of the Quick start instructions, we have provided the contact information of the first author, who is well-experienced and motivated to support the users and implement new options based on their request. This is something we have done routinely also for the other web-tools implemented in the group¹. As an example, the same group of authors has developed and maintained a popular web-app, SynergyFinder, and its second version that implemented requests from its active user group (>15000 users, >350 citations since 2017).

lanevski A, Giri AK, Aittokallio T. SynergyFinder 2.0: visual analytics of multi-drug combination synergies. *Nucleic Acids Res.* 2020 Jul 2;48(W1):W488-W493. doi: 10.1093/nar/gkaa216. PMID: 32246720.

lanevski A, He L, Aittokallio T, Tang J. SynergyFinder: a web application for analyzing drug combination dose-response matrix data. *Bioinformatics.* 2017 Aug 1;33(15):2413-2415. doi: 10.1093/bioinformatics/btx162. PMID: 28379339.

¹ <https://www2.helsinki.fi/en/researchgroups/computational-systems-medicine/software-tools-0>

Reviewer #2 (Expertise: scRNASeq for the analysis haematopoietic datasets):

The manuscript by lanevsky et al. describes ScType, a new platform for the analysis of single-cell RNAseq data sets available as an interactive web tool and R source code. Its main features are a novel tool for the unsupervised, marker-based annotation of the identified cell clusters and the implementation of specific methods for analyzing cancer-related single-cell experiments.

Annotation of cell clusters is performed through a scoring function in which normalized marker-genes expression values are multiplied by a novel, tissue-specific, marker-gene cell-specificity index. ScType uses a marker-genes database obtained by combining 2 of the most significant resources currently available in the literature (CellMarker and PanglaoDB) with the addition of cell-type markers derived from the manual curation of recent publications aimed at covering tissues underrepresented in the original version of the database. Clusters annotation is a common and essential step of single-cell transcriptomic data analysis, and several methods have been proposed to perform this task. ScType outperformed most alternative approaches in annotating clusters with the appropriate cell-type labels in a comparative study based on published experimental data sets. In cases where ScType cannot provide a significant gain in accuracy, it gives the results in a considerably shorter amount of time.

In single-cell RNA data sets involving cancer-related samples, ScType offers the possibility to identify which clusters are enriched in malignant/healthy cells by simultaneously analyzing the ScType_SNV score distribution and the results generated by CopyKAT. ScType_SNV score is an index that measures how mutations in the cancer-consensus-gene list are distributed among the cells and clusters. CopyKAT instead classifies as aneuploid (malignant) or diploid (healthy) individual cells using a Bayesian segmentation approach. As an example of analysis performed on a cancer data set, authors re-analyses the data from one of their previous publications regarding a sample isolated from an AML patient.

Major comments

1. Contrary to many alternative methods, ScType takes advantage of both positive and negative markers for annotating cell clusters. However, it is known that, for both biological and technical reasons, many genes considered as markers are in reality detected (non-zero) only in a fraction of the cells genuinely belonging to a specific cell type [Guo, H., & Li, J. (2021)]. Since the percentage of cells with missing expression can be significant (up to 80%), this can significantly impact smaller clusters.

We agree with the reviewer that the dropout rates can be significant in single-cell data (up to 80%), which may notably impact the clustering solutions and annotation results. To further investigate this effect, we utilized the Splatter method (Zappia, L et al. 2017, as suggested by the reviewer below in comment 3), and generated 45 simulated datasets with various dropout rates: 15 datasets with ~50% dropouts, 15 datasets with ~65% dropouts, and 15 datasets with ~80% dropouts (dropout.shape=-0.5 and dropout.mid -2.5, 0, 2.5, respectively). Each simulated dataset contained 10 cell types with varying cell proportions, as defined by the scaled lognormal random deviates (e.g. 42.3%, 22.7%, 12.6%, 4.5%, 4.5%, 4.1%, 4.0%, 2.9%, 1.8%, 0.3%). Each dataset had "Moderate" similarity among the cell types (i.e., de.facScale parameter was randomly set to either 0.5, 0.6 or 0.7). In addition, for each dataset, we

extracted marker genes from the simulated reference data by performing differential expression analysis using the Wilcoxon rank sum test, similarly to the recent comparative evaluation work:

Huang Q, Liu Y, Du Y, Garmire LX. Evaluation of Cell Type Annotation R Packages on Single-cell RNA-seq Data. *Genomics Proteomics Bioinformatics* 2020. Dec 23:S1672-0229(20)30144-3. doi: 10.1016/j.gpb.2020.07.004. PMID: 33359678.

For each cell type, we generated both highly-specific markers (ten top-ranked marker genes separately for each cell type), identified through differential expression, as well as low-specificity markers (ten top-ranked marker genes for a mixture of all simulated cell types including the cell type in question, see Methods pages 19-20), which were used as the input for the ScType and other methods.

Using the simulated datasets, we compared the performance and running time of ScType, scSorter, and SCINA methods (please see Supplementary Figure 2 below). We note that scCATCH method was not used in the comparison, since its implementation does not allow the use of custom markers (this issue is discussed in more detail in the scCATCH GitHub site: <https://github.com/ZJUFanLab/scCATCH/issues/31>). As a result, we found that ScType was the most accurate annotation method in the simulated datasets with approx. 50% dropout rate, and it also showed comparable performance with the other methods in datasets with higher dropout rates, while being again the fastest method in all the datasets ($P < 0.0001$).

Supplementary Figure 2. Comparison of ScType, scSorter and SCINA accuracy and runtime performance. (a) The cell type annotation accuracy was tested in 45 simulated datasets with various dropout rates. The horizontal lines indicate the median, the boxes the interquartile range (IQR), and the whiskers are $Q1 - 1.5 \times IQR$ and $Q3 + 1.5 \times IQR$ (where $Q1$ and $Q3$ are firsts and third quartiles), across the 15 simulated datasets in each condition. (b) The ScType running time was faster compared to the other methods ($P < 0.0001$, Wilcoxon test). *Note:* the time-axis is log-scaled.

These results show that the cell type annotation accuracy of ScType is similarly impacted by the dropout rate, compared to the other methods, as was expected, but even at the highest drop-out rate the accuracy does not go below 90%. We have now described these new results in the revised manuscript (page 6):

Evaluation of dropout effects and unknown cell types in simulated scRNA-seq data

The dropout rates can be significant in single-cell data (up to 80%), which may notably impact the clustering solutions and annotation results. To investigate this effect, we utilized the Splatter method³⁵, and generated 45 simulated datasets with various dropout rates: 15 datasets with ~50% dropouts, 15 datasets with ~65% dropouts, and 15 datasets with ~80% dropouts (see Methods). For each cell type, we generated both highly-specific markers (ten top-ranked marker genes separately for each cell type), and low-specificity markers (ten top-ranked marker genes for a mixture of cell types including the cell type in question), which were used as the input for the ScType and the other methods (*note*: scCATCH was not used in this comparison since its implementation does not allow for the use of custom markers). The comparison results in the simulated datasets showed that the ScType annotations are relatively robust against high percentages of cells with missing expression, with the annotation accuracy remaining above 90% even at 80% dropout rate, generally outperforming the other methods both in terms of the annotation accuracy and running time (Fig. S2).

2. Despite being a standard step in many single-cell RNA analysis workflows, the z-score transformation is suboptimal when dealing with single-cell transcriptomic data. I am aware that this step might be difficult to replace with alternative procedures, but it represents a limitation worth mentioning if not thoroughly investigated.

As noted by the reviewer, z-score is a standard step in many single-cell RNA analysis workflows despite being sometimes suboptimal. We have used z-core as it allows

- (1) Normalization of the gene expression measurements for each gene across cells
- (2) Calculation of an appropriate marker expression score that combines multiple markers and is comparable across different cell types
- (3) It is fast to compute and is computationally efficient

However, in order to investigate the impact of z-score transformation on the performance of ScType annotations, we utilized the same 45 simulated datasets (see response to comment 1 above), and ran ScType with and without the z-score transformation (see Supplementary Figure 5 below). As a result, we observed a drastic decrease in the annotation accuracy after removing the z-score transformation step (82.0% vs 97.2% of correctly assigned cells; $P < 0.001$, Wilcoxon test). We have now discussed the importance of using the z-score transformation briefly in the discussion (page 14):

In addition, ScType utilizes z-score transformation for computationally-efficient combination of multiple markers, which may be suboptimal when dealing with single-cell transcriptomic data. However, our results in the simulated datasets showed a drastic decrease in the annotation accuracy when removing the z-score transformation ($P < 0.001$, Wilcoxon test; Fig. S5).

Supplementary Figure 5. The impact of z-score transformation on the ScType annotation accuracy in the 45 simulated datasets with various dropout rates ($P < 0.001$, Wilcoxon test). The horizontal lines indicate the median, the boxes the interquartile range (IQR), and the whiskers are $Q1 - 1.5 \cdot IQR$ and $Q3 + 1.5 \cdot IQR$ (where $Q1$ and $Q3$ are firsts and third quartiles), across the 45 simulated datasets in each condition.

3. In the manuscript's main text, it is mentioned that ScType label as “unknown” a cluster corresponding to a cell type not included in the reference database. Still, it is not explained which criteria have been used for this “missing” annotation.

We thank the reviewer for pointing this out and do apologize for not explaining how the “unknown” cell types were annotated in the submitted version. We have now added the below sentence to the revised manuscript (pages 18-19):

We consider a low ScType score (less than quarter the number of cells in a cluster), or a negative ScType score to indicate a low-confidence cell-type annotation, which are assigned as “unknown” cell type in the web-tool.

Given the limited amount of information still available nowadays about the markers expressed by many cell types, correctly identifying “unknown” cell types is crucial and has to be stated clearly how this is done.

We totally agree. In order to investigate the impact of “unknown” cell types on the performance of ScType, scSorter and SCINA methods, we utilized again the same 45 simulated datasets and used a cross-validation scheme of “leave-1-cell-type-out” to evaluate the accuracy of identifying the unknown cell groups. More specifically, we removed the specific marker signatures of one of the cell types in each of the 45 datasets, and then performed cell type annotations with all the methods. We observed that ScType and SCINA were able to correctly assign “unknown” cell types in most of the datasets, 43 out of 45 datasets (95.5%) and 41 out of 45 datasets (91.1%), respectively, while ScSorter correctly identified “unknown” cell types only in 22 out of 45 datasets (48.8%). We have incorporated these new analysis in the revised manuscript (pages 6-7):

Due to the limited amount of information currently available on the markers expressed in many of the cell types, accurate identification of “unknown” cell types is an important practical task. To investigate the impact of unknown cell types on the performance of ScType, scSorter and SCINA methods, we utilized the same 45 simulated datasets and used a cross-validation scheme of “leave-one-cell-type-out” to evaluate the accuracy of identifying the unknown cell groups. More specifically, we removed the specific marker signatures of one of the cell types at a time in each of the 45 datasets, and then performed the cell type annotations with all the methods. In ScType, we considered a low ScType score (less than quarter the number of cells in a cluster) or a negative ScType score as low-confidence cell type annotations, which were assigned as “unknown” cell types. We observed that ScType and SCINA were able to correctly assign unknown cell types in most of the simulated datasets, 43 out of 45 datasets (95.5%) and 41 out of 45 datasets (91.1%), respectively, while ScSorter correctly identified unknown cell types only in 22 out of 45 datasets (48.8%).

To annotate the “unknown” cell types lacking specific markers in our database, ScType performs differential expression analysis between unknown cell cluster vs. all the other clusters to identify up- and down- regulated genes in an “unknown” cluster. These markers can be used for further investigation of “unknown” cell types.

To address issues 1,2 and 3 aforementioned, I would suggest running a comparative study based on data generated with a method designed to mimic single cell transcriptional profiles [Zappia, L et al. (2017); Zhang, X., Xu, C., & Yosef, N. (2019), Dibaieinia, P., & Sinha, S. (2020)]. The data set could be composed of clusters corresponding to a variable number of cell types with known and unknown marker genes. This would allow to:

- 1. Compare ScType to alternative methods under controlled scenarios.*
- 2. Investigate the impact of negative markers on minor cluster annotation.*
- 3. Measure how z-score transformation affects cell type identification.*
- 4. Test the performance of ScType in detecting “unknown” cell types.*

We thank the reviewer for suggesting a Splatter method (Zappia, L et al. 2017), which we have used to generate 45 simulated datasets (please see our responses to comments 1-3 above). We hope these additional analyses address the above points adequately. Please also see our response to the first comment of reviewer 1 regarding the use and benefits of negative markers for annotating sub-groups of closely-related cell types.

4. The ScType web tool is well designed, intuitive, and responsive. The platform runs smoothly using the default data set provided by the authors as examples. When I tried to analyze my data set, I uploaded the data (count matrix of 8K human cells) and performed the 2 quality control steps (filtering of cells and selecting variable genes). However, the remaining and most relevant parts of the workflow did not work. I tried multiple operating systems and browsers with no success.

We deeply apologize for the inconvenience caused. There was indeed a problem caused by the automated update of the PHP version, that we noticed only after the submission. We have now disabled all the automated updates, ensuring this problem will not appear anymore. We have now carefully tested that everything is fixed and working properly. We have also asked our colleagues in various sites to test the proper operation of the web-portal and the R-code and to reproduce the same results based on the step-by-step tour that we provided

now for the web-tool usage. In total, there have been so far 5 in-house beta-users and 4 outside users of ScType, and they have been able to analyze their own datasets indicating the robust functioning of the webtool.

Minor comments

1. In general, I felt the first part of the manuscript focused on the unsupervised cluster annotation and the second, more cancer-oriented, a bit unrelated. Both are interesting and promising, but I think they might have a slightly different audience.

We agree that the two parts of the manuscript have different focus, the first one solving a more general cell type identification problem, while the second one showing how the cell type information can be used to detect malignant and non-malignant cell populations for cancer applications. The latter focus comes from our own use of the ScType method at FIMM to design effective and less-toxic anticancer combinations that selectively co-inhibit mainly the malignant cell types, while avoiding inhibition of nonmalignant cells, thereby increasing their likelihood for clinical translation. We feel that keeping the cancer application in the work makes it more exciting for the community, since it shows in a relevant case study how to use ScType for biomedical applications. We have now made this connection between the general-purpose tool and cancer applications clearer in the revised manuscript (pages 3 and 10).

2. A simple toy example showing how the cell-type specificity-score and the enrichment-score are calculated on a small set of marker genes would make it easier for readers to have an intuition of how ScType annotation works.

We have now illustrated the method for calculating the cell-type specificity-score and the enrichment-score using a toy example (please see the new Supplementary Figure 3 below).

a

b

Supplementary Figure 3. An example of calculating marker specificity score and ScType score.

(a) In this toy example, Gene1 is a non-specific marker, and expressed in all 3 cell types (T1,T2,T3), whereas Gene2, Gene3, and Gene5 are specific markers for cell types T1, T2, and T3, respectively. Gene4 is a positive marker for cell type T3 and negative marker for cell type T1. Gene 6 is a positive marker for cell types T2 and T3. In order to calculate the marker specificity score (the right column), the marker occurrence across the cell types (middle column) is counted and scaled between 0 to 1 (0 indicates non-specific markers, maximum occurrence; and 1 for highly specific markers, minimum occurrence). **(b)** To calculate the enrichment score based on the marker expression in ScType, the raw expression data is first normalized and Z-transformed. Next, the normalized matrix is multiplied by the cell-type specificity score. Then, the expression scores of all the positive markers corresponding to a particular cell type are summarized into a single enrichment score by summing them and dividing by square root of their number. The same is done for the negative markers. Finally, the negative marker expression score is subtracted from the positive score to obtain the final enrichment score. Individual cells are assigned to a cell type based on the maximum value for the cell type marker set.

In addition, we implemented a bubble plot showing all the cell types that were considered by ScType for cluster annotation (see the figure below). The outer (grey) bubbles correspond to each cluster (the bigger the bubble, the more cells in the cluster), while the inner bubbles correspond to the considered cell types for each cluster, with the biggest bubble corresponding

to the assigned cell type. This feature is also describe in the Quick start instructions in the Github (see <https://github.com/lanevskiAleksandr/sc-type/blob/master/README.md#bubble>)

A bubble plot showing all the cell types that were considered by ScType for the cell type cluster annotation

3. It is not clear whether the experimental data sets used in the comparative study have also been used to define cell type marker-gene included in the database. Clarify this point since it is relevant in the evaluation of the performance of ScType.

Only the human brain data had been used to define the markers in the marker gene database based on the publication list found in the PangloDB paper's reference list and website (<https://panglaodb.se/papers.html>). However, we excluded the markers identified from the human brain data while annotating the cell cluster in the comparative study to make the comparison unbiased. The other marker databases used in ScType database, CellMarker database, used more than 100 000 publications to identify the markers. However, it did not list the publications used for defining its marker genes. Hence, we could not confirm whether the CellMarker database have incorporated markers from the experimental dataset used in the comparative study. We have now clarified this in the revised manuscript (page 19).

REVIEWERS' COMMENTS

Reviewer #1 (Remarks to the Author):

All my previous comments have been answered appropriately.

I have no further comments.

Reviewer #2 (Remarks to the Author):

The authors performed an extensive simulation study to verify how ScType behaves in different conditions. The impact of drop-out rates of varying intensity has been tested. The proposal outperformed alternative methods in terms of accuracy in the majority of the settings while requiring a lower computational running time.

ScType also demonstrated a better capacity in identifying unknown cell types with respect to ScSorter and slight improvement with respect to SCINA.

The authors provide quantitative justification for applying the z-score transformation to the gene expression data by showing a significant improvement of the method's accuracy.

The platform's web version now works with both the pre-loaded and the user uploaded data. I would suggest the authors verify if there is any limitation in input data set size and whether these limits change according to the input format since I encountered some issues uploading an experiment with more 22K cells in csv format (~1.3 Gb). Also, if possible, I would add the possibility of changing the prefix for discriminating mitochondria gene names (mt-,mt_, mt.) in the ScType web version.

The step-by-step calculation of the ScType score reported in Supplementary Fig. 3 helps the reader understand the framework. There might be a minor typo in the "Positive marker set expression" second-row name.

REVIEWERS' COMMENTS

Reviewer #1 (Remarks to the Author):

All my previous comments have been answered appropriately.
I have no further comments.

We thank the reviewer for the constructive comments on our work.

Reviewer #2 (Remarks to the Author):

The authors performed an extensive simulation study to verify how ScType behaves in different conditions. The impact of drop-out rates of varying intensity has been tested. The proposal outperformed alternative methods in terms of accuracy in the majority of the settings while requiring a lower computational running time.

ScType also demonstrated a better capacity in identifying unknown cell types with respect to ScSorter and slight improvement with respect to SCINA.

The authors provide quantitative justification for applying the z-score transformation to the gene expression data by showing a significant improvement of the method's accuracy.

We thank the reviewer for the positive comments on the revised work.

The platform's web version now works with both the pre-loaded and the user uploaded data. I would suggest the authors verify if there is any limitation in input data set size and whether these limits change according to the input format since I encountered some issues uploading an experiment with more 22K cells in csv format (~1.3 Gb).

We have now added a note to documentation that the maximum input file size is 1Gb for the web-version, please see https://sctype.app/sctype_docs/index.html#features. This max file size is due to the current 6Tb space reserved for the server machine, and because 1Gb dataset report folder would occupy ~10 Gb disk space, this will enable maximum 600 reports of such size from different users. However, the input files larger than 1Gb can be processed by the users using ScType R-package <https://github.com/lanevskiAleksandr/sc-type>.

Also, if possible, I would add the possibility of changing the prefix for discriminating mitochondria gene names (mt-,mt_, mt.) in the ScType web version.

We have now added the possibility of changing the prefix for discriminating mitochondria gene names (mt-,mt_, mt.) in the ScType web-version. ScType web-app now accepts mt-, mt_, and mt. prefixes and it automatically identifies the prefix used, which is now also mentioned in the documentation, please see https://sctype.app/sctype_docs/index.html#features.

The step-by-step calculation of the ScType score reported in Supplementary Fig. 3 helps the reader understand the framework. There might be a minor typo in the "Positive marker set expression" second-row name.

The typo has been corrected, thank you for pointing this out.